# Anomalous thickness dependence of the vortex pearl length in few-layer NbSe$_2$

Nofar Fridman [1,2] ✉, Tomer Daniel Feld[1,2], Avia Noah [1,2,3], Ayelet Zalic [1,2], Maya Markman[1,2], T. R. Devidas [1,2], Yishay Zur[1,2], Einav Grynszpan[1,2], Alon Gutfreund [1,2], Itai Keren [1,2], Atzmon Vakahi[2], Sergei Remennik [2], Kenji Watanabe [4], Takashi Taniguchi [5], Martin Emile Huber [6], Igor Aleiner[7], Hadar Steinberg[1,2], Oded Agam [1] ✉ & Yonathan Anahory [1,2] ✉

The coexistence of multiple types of orders is a common thread in condensed matter physics and unconventional superconductors. The nature of superconducting orders may be unveiled by analyzing local perturbations such as vortices. For thin films, the vortex magnetic profile is characterized by the Pearl-length $\Lambda$, which is inversely proportional to the 2D superfluid density; hence, normally, also inversely proportional to the film thickness, $d$. Here we employ the scanning SQUID-on-tip microscopy to measure $\Lambda$ in NbSe$_2$ flakes with thicknesses ranging from $N = 3$ to 53 layers. For $N > 10$, we find the expected dependence $\Lambda \propto 1/d$. However, six-layer films show a sharp increase of $\Lambda$ deviating by a factor of three from the expected value. This value remains fixed for $N = 3$ to 6. This unexpected behavior suggests the competition between two orders; one residing only on the first and last layers of the film while the other prevails in all layers.

Superconductivity in the presence of competing or intertwined order parameters has generated much interest over the last decades. Detecting the presence of an additional order parameter and determining its influence on superconductivity is crucial in unveiling the pairing mechanism. In particular, order-parameter competition could involve two superconducting order parameters with different pairing symmetries[1] or a single superconducting channel and an order parameter related to another degree of freedom, such as charge density wave[2,3] or spin density wave[4].

Although usually discussed in the context of high-temperature superconductors[5,6], competing orders are also relevant to many other layered materials, such as twisted bilayer graphene[7] and NbSe$_2$[1]. Moreover, in the case of thin films with few atomic layers, the competition between distinct order parameters residing on the surface and in the bulk also becomes a possibility. This is due to the disparity in properties exhibited by the surface and bulk regions, notably exemplified by the presence of Rashba spin-orbit coupling on the surface[8]. However, as far as we are aware, the examination of competition between surface and bulk order parameters, both experimentally and theoretically, has not been documented in prior studies. In this work, such a competition is uncovered through measurements of the magnetic field profile of superconducting vortices in NbSe$_2$. NbSe$_2$ is a layered superconductor which sustains superconductivity with $T_c > 4.2$ K for any thickness above 3 layers, and thus is uniquely suitable for such an experiment.

Superconducting vortices are often exploited as local perturbations used to probe the properties of the order parameter[9–12]. Vortices consist of a core where the superconductivity is locally suppressed on the scale of the coherence length $\xi$ where the magnetic field can penetrate. This field is screened by the surrounding superconducting

[1]The Racah Institute of Physics, The Hebrew University, Jerusalem, Israel. [2]Center for Nanoscience and Nanotechnology, Hebrew University of Jerusalem, Jerusalem, Israel. [3]Faculty of Engineering, Ruppin Academic Center, Emek-Hefer 40250 Monash, Israel. [4]Research Center for Electronic and Optical Materials, National Institute for Materials Science, 1-1 Namiki, Tsukuba, Japan. [5]Research Center for Materials Nanoarchitectonics, National Institute for Materials Science, 1-1 Namiki, Tsukuba, Japan. [6]Departments of Physics and Electrical Engineering, University of Colorado Denver, Denver, CO, USA. [7]Google Quantum AI, Santa Barbara, CA, USA. ✉e-mail: nofarfri.friedman@mail.huji.ac.il; agam.oded@gmail.com; yonathan.anahory@mail.huji.ac.il

currents and results in magnetic flux quantization. In bulk superconductors, the magnetic field is screened exponentially with a characteristic scale known as the London penetration depth $\lambda_L$. By contrast, for a film of thickness $d < \lambda_L$, screening is less effective and is governed by the Pearl length $\Lambda = 2\lambda_L^2/d$ [13–15]. In this limit, the magnetic field decays as $1/\Lambda r$ near the vortex core and as $\Lambda/r^3$ for distances greater than $\Lambda$, where $r$ is the distance from the vortex's center. Therefore, near the vortex core, there is no characteristic length scale for the magnetic field screening. In this work, we measure the Pearl length, which is a fundamental characterizer of superconductors that is highly sensitive to variation of the two-dimensional superfluid density. This sensitivity is particularly important when studying phase transitions that involve changes in the superconducting order, as such transitions naturally alter the superfluid density.

We have measured the thickness dependence of the Pearl length in NbSe$_2$ flakes with thicknesses ranging from $N = 3$ to 53 layers. For this purpose we employ a highly sensitive microscopy technique of SQUID-on-tip [16,17] (SOT, see Fig. 1) coupled to a tuning fork designed to measure gradients in minute magnetic signals emitted by a vortex, particularly in cases where the Pearl length significantly exceeds the size of micron-scale flakes. Our data presents the anticipated $1/d$ dependence for flakes of thicknesses $N \gtrsim 10$ layers. However, strikingly, $\Lambda$ largely deviates from the expected $1/d$ dependence for thinner films. Such deviation has not been previously reported in NbSe$_2$ neither in transport [18] nor in tunneling [3,9,19–21] studies. We suggest that the sharp jump in $\Lambda$ can be attributed to the competition between bulk and surface superconductivity.

## Results

NbSe$_2$ was mechanically exfoliated to obtain flakes with thicknesses ranging from $N = 3$ to 53 layers ($d = 1.9$ to 33 nm). To prevent sample degradation, thin flakes with $N \le 14$ layers were encapsulated with h-BN from both sides (see Methods). Figure 2a, b depict representative optical images of thin films, showcasing the presence of atomically flat terraces. The number of layers, which corresponds to the thickness, is indicated for each observed area. For thin flakes ($N \le 14$ layers), sample thickness was measured using cross-sectional scanning transmission electron microscopy (STEM), as described in the Methods and shown in Supplementary Fig. 1. For thicker samples with $N > 14$ layers, the

thickness was measured using an atomic force microscope (AFM). Energy-dispersive X-ray spectroscopy (EDS) analysis was performed on the flakes ($N \le 14$ layers) and showed no surface contamination (Supplementary Fig. 2).

The samples are mounted in a scanning SOT microscope to conduct magnetic imaging. In particular, we image the out-of-plane component of the magnetic field $B_z(h, \mathbf{r})$ at the surface of superconducting NbSe$_2$ in the presence of vortices at 4.2 K (Fig. 1), where $h$ is the distance of the tip from the surface and $\mathbf{r} = (x, y)$ is the in-plane coordinate. Vortices are visually identified as bright regions in the acquired images (see Figs. 1 and 2c). To ensure non-overlapping signals from individual vortices, we adjusted the external magnetic field $\mu_0 H_z \sim 1$ mT, that controls the vortex density. The positions of the vortices measured at different magnetic fields are marked with orange dots in Fig. 2a, b (see also Supplementary Fig. 3).

We commence by clarifying our capacity to extract the Pearl length even in cases where it considerably surpasses the flake size. The magnetic field profile of the Pearl vortex exhibits a characteristic gradual transition from the short-distance asymptotic behavior, described by $1/\Lambda r$, when $h \ll r \ll \Lambda$ to the long-distance behavior of $\Lambda/r^3$ when $r \gg \Lambda$. Examples of such profiles are presented in Fig. 2d for $h = 260$ nm with $\Lambda = 1.5$ μm and $\Lambda = 100$ μm and plotted for a typical SOT field of view ($|r| \le 1.1$ μm). For this region of interest, $h \ll r \ll \Lambda$, both profiles are governed by the same power-law decay. Therefore, the two profiles differ solely by their magnitude, which is inversely proportional to $\Lambda$. The slow $1/r$ decay implies that magnetic fields resulting from neighboring vortices and the Meissner current associated with the sample's edges add up to an approximately constant background. This background can be effectively eliminated by measuring the gradient of the signal yielding a localized signal from which the Pearl length can be extracted with an error of order $(h/L)^2$, where $L$ is the typical size of the flake or the distance to a neighboring vortex (see Supplementary Note 1). As we typically measure at $h \lesssim 0.4$ μm and $L$ is typically larger than 1 μm, this error is small. To substantiate this result, we fit simulated isolated vortices and compare them with fits for vortices surrounded by randomly distributed vortices at a typical distance we encounter in experimental conditions ($\sim 2$ μm). Our results, presented in Supplementary Fig. 4, show that the influence of these additional vortices on $\Lambda$ is up to 10%. Thus, when the magnetic field is sufficiently weak, it is feasible to measure Pearl lengths in the range of a few hundred microns, even with a limited field of view of just a few microns. The capability to measure screening lengths of such magnitude is essential for the conclusions drawn in this work.

To measure the gradient of the out-of-plane component of the magnetic field, $B_z(h, \mathbf{r})$, we mechanically couple the SOT to a tuning fork, as shown in Fig. 2f. The tuning fork is set to oscillate, resulting in a periodic lateral motion of the SQUID loop at a frequency of approximately 32 kHz, as indicated by the blue double-headed arrow in the inset of the Figure. When the amplitude of the tuning fork oscillation is small, the in-phase ac signal, $B_z^{ac}(h, \mathbf{r})$, is proportional to the gradient of the corresponding dc signal along a chosen direction. Setting the $x$ axis along this direction, $B_z^{ac}(h, \mathbf{r}) \cong x_{ac} dB_z(h, \mathbf{r})/dx$, where $x_{ac}$ is the oscillation amplitude of the SQUID loop. Using an independent measurement of the oscillation amplitude (see Methods and Supplementary Note 2), we are able to deduce the spatial derivative of the static signal $B_z(h, \mathbf{r})$, shown in Fig. 2e. Typically we set $x_{ac} \lesssim 100$ nm, which yields a large signal while keeping the following approximation $B_z^{ac}(h, \mathbf{r}) \cong x_{ac} dB_z(h, \mathbf{r})/dx$ within our experimental uncertainty. Moreover, by conducting measurements at a frequency of 32 kHz, we are able to effectively eliminate the $1/f$ noise, which is typically at least two orders of magnitude higher below 1 kHz [16,17].

Figure 3a–d display typical images of the spatial derivative along the $x$ axis of the out-of-plane component of the magnetic field ($B_z^{ac}(h, \mathbf{r})/x_{ac}$) measured for various thicknesses ($N = 3, 6, 7,$ and 14 layers). Figure 3e–h show the best fits achieved for each image to the

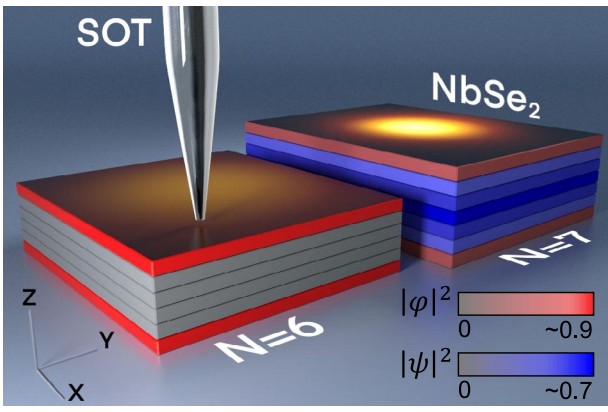

**Fig. 1 | Schematic diagram of the experimental setup.** Two NbSe$_2$ flakes with $N = 6$ and 7 layers. SQUID-on-tip (SOT) magnetic images of vortices representing the magnitude of the out-of-plane component of the magnetic field $B_z(h, \mathbf{r})$ are overlaid on the top surface, where $h$ is the distance of the tip from the surface and $\mathbf{r} = (x, y)$ is the in-plane coordinate. The contour of each layer is colored according to the superconducting order parameters: $\psi$ (blue) residing in all layers, and $\varphi$ (red) is a different superconducting order parameter confined to the surface. The color intensity encodes the amplitude of each order parameter, where gray represent the normal state $|\psi| = 0$. For $N = 7$, the two order parameters are finite, while for $N = 6$, $|\psi| = 0$, while $|\varphi|$ remains finite and confined to the first and last layer.

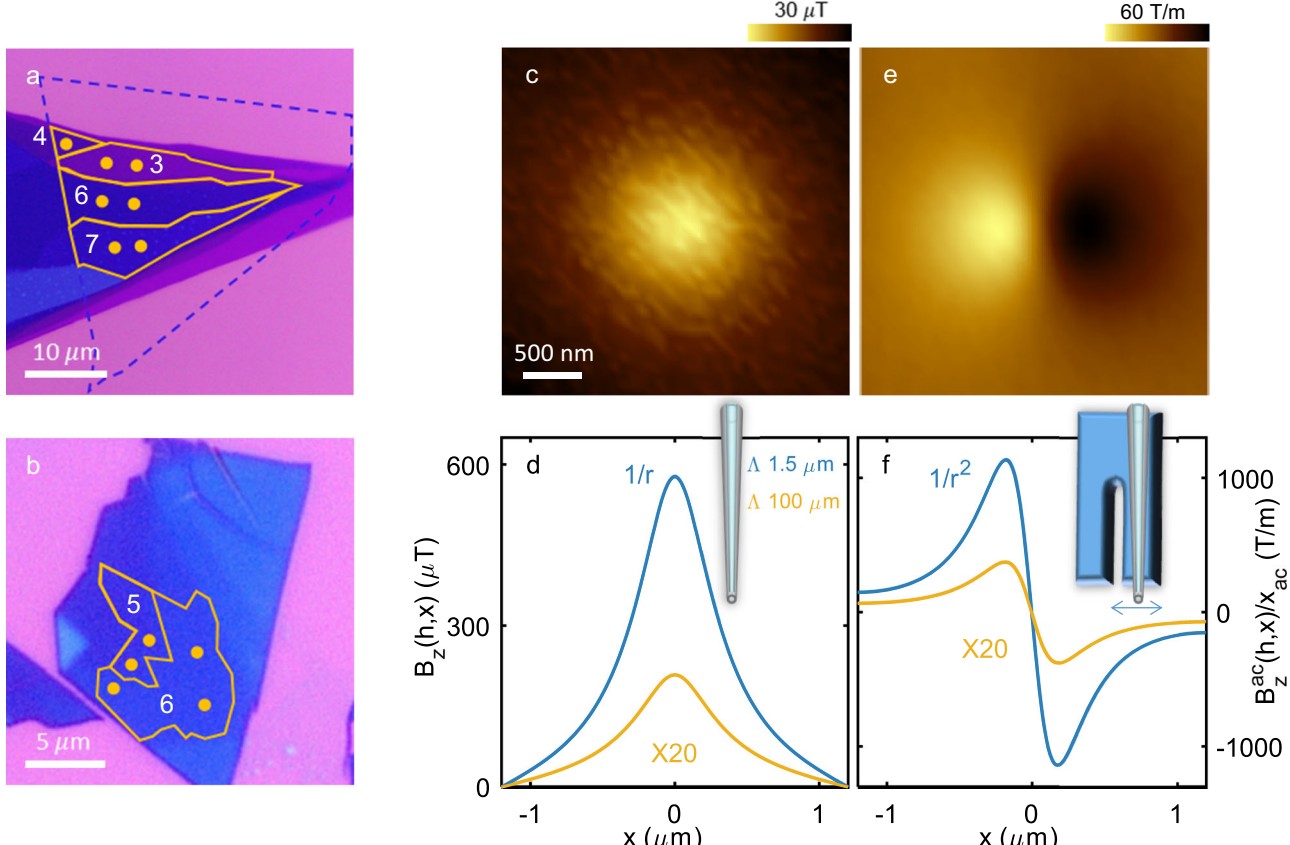

**Fig. 2 | Pearl model and SQUID-on-tip images of NbSe₂ at 4.2 K. a, b** Optical images of the ultra-thin flakes and the locations of the imaged vortices. The numbers indicate the number of layers $N$, and orange lines outline the terraces' edges. **a** The blue dashed line shows the area covered with top and bottom hBN. **b** The entire field of view is encapsulated with top and bottom hBN (**c**) SQUID-on-tip (SOT) image of the out-of-plane component of the magnetic field $B_z(h,\boldsymbol{r})$ of an isolated vortex located in the $N=7$ layer region shown in panel a, where $h$ is the tip-to-surface distance and $\boldsymbol{r}=(x,y)$ is the in-plane coordinate. **d** Calculated magnetic profile $B_z(h,x)$ of vortices using the Pearl model with Pearl length $\Lambda=1.5$ (blue) and 100 μm (yellow). Both profiles decay with the same power law ($1/\Lambda r$) for $r \ll \Lambda$. (inset) Illustration of a SOT scanning the surface monotonically as opposed to (**f**). **e** The same vortex as in **c**, measured while the tip oscillates along the $x$ axis. The image shows the field component oscillating in phase with the SQUID loop $B_z^{ac}(h,\boldsymbol{r})$ divided by the motion amplitude $x_{ac}$ resulting in spatial derivative of the image shown in (**c**) along the $x$ axis. **f** Same as (**d**) but showing the spatial derivative along the $x$ axis $B_z^{ac}(h,x)/x_{ac}$. (inset) Illustration of the SOT coupled to a tuning fork oscillating along the $x$ axis (blue double-headed arrow).

Pearl model. This model requires two parameters, the height $h$ and the Pearl length $\Lambda$. We determine $h$ by sensing the sample surface with the tuning fork, as in an AFM, then retract a known amount, leaving $\Lambda$ as our sole fitting parameter. The uncertainty on the measurement of $h$ is 15 nm, which propagates to an uncertainty in the Pearl length of 8–10% (Supplementary Note 3). In addition, we take into account our finite SOT resolution by convoluting the modeled image with a circle corresponding to the SQUID's diameter, which we determine by measuring the field period of the critical current oscillations[16,17]. To demonstrate the good agreement between fits and measurements, we show the cross-sections of derivatives of the model and the $B_z^{ac}(h,\boldsymbol{r})/x_{ac}$ signal along the $x$ axis of the image (Fig. 3i–l). From these fits we obtain $\Lambda=111,101,30$, and 12 μm for $N=3,6,7$, and 14, respectively. Notably, the measured signal magnitude is approximately twice as large for $N=6$ compared to $N=3$, despite obtaining similar $\Lambda$ values for both cases. The difference of the signals is attributed to the difference in the tip height at which the measurement was conducted: $h=360\pm15$ and $260\pm15$ nm for $N=3$ and 6, respectively. The nearly constant value of $\Lambda$ for two different thicknesses manifests a puzzling deviation from the expected $1/d$ dependence.

Figure 4 summarizes the measured values of the Pearl length $\Lambda$ as a function of the thickness $d$ ranging from $N=3$ to 53 layers plotted on a logarithmic scale. According to the Pearl model, this plot is expected to exhibit a slope of $-1$ (representing the $d^{-1}$ dependence of the Pearl

length) along with an offset determined by the London penetration depth $\lambda_L$. Indeed, for $N\gtrsim10$ layers, such dependence is observed, enabling us to estimate $\lambda_L=230$ nm in good agreement with bulk value $\lambda_L=200$ nm measured at 4.2 K[22–25]. Notice that, for thicker films, the thickness was measured with AFM, and the flakes were not encapsulated. Consequently, the uncertainty on the thickness is larger, which explains partially the scatter of the data points around $1/d$.

For $N=6$, we observe a drastic increase in $\Lambda$, to 101 μm, which is three times larger than the expected value of 28 μm according to Pearl model. Moreover, $\Lambda$ remains surprisingly constant, within $100\pm15$ μm, for $N=3,4,5$, and 6 layers. However, take note that in a different sample with $N=6$, we obtained a significantly different value of $\Lambda=39$ μm. This value is somewhat closer to the expected value from Pearl's model of 28 μm. These findings suggest that near a critical thickness of $N=6$, where the system undergoes a phase transition, finite size effects are significant.

Two key features characterize the measured Pearl length dependence on the film thickness. The constant value of $\Lambda$ below the critical thickness, and the sharp jump near $N=6$. In what follows, we discuss potential experimental issues, and explain why their impact on our findings is negligible. Consider first the issue of sensitivity limitations. In Fig. 3i, we compare the profile of the image measured in Fig. 3a with three simulated profiles. The best fit (in yellow) and two other profiles obtained by changing $\Lambda$ by $\pm15\%$ (blue and red). The figure

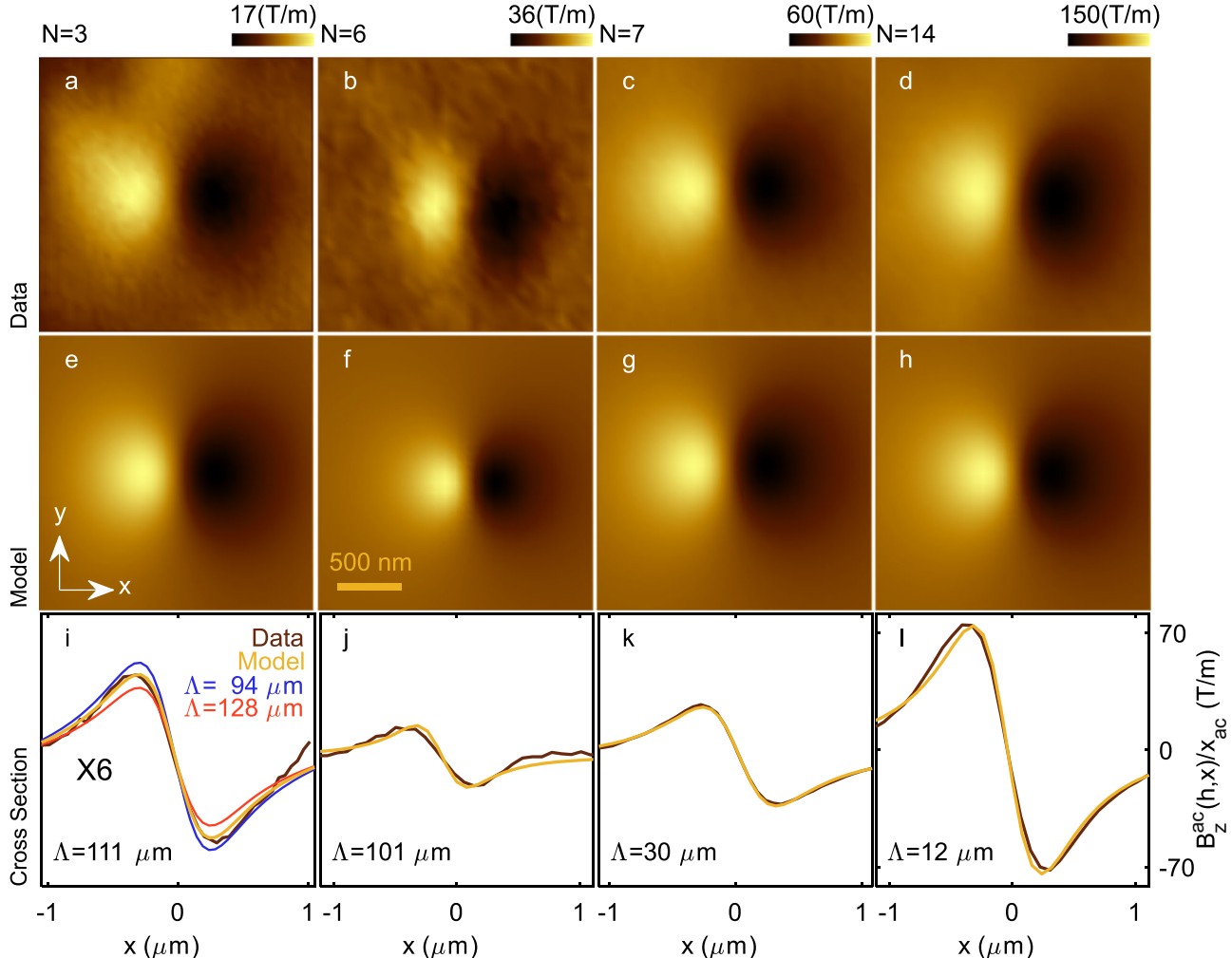

**Fig. 3 | Images of the magnetic field gradient near vortices in samples of distinct thicknesses. a–d** Spatial derivative along the $x$ axis of the out-of-plane component of magnetic field $B_z^{ac}(h, \mathbf{r})/x_{ac}$ of a vortex located in a region of $N = 3$ **a**, 6 **b**, 7 **c**, and 14 **d** layers, where $h$ is the tip-to-surface distance, $\mathbf{r} = (x, y)$ is the in-plane coordinate, and $x_{ac}$ is the motion amplitude of the SQUID loop. Images were acquired from $h = 360$ **a**, 260 **b** 360 **c**, and 360 **d** nm (**e–h**) Calculated theoretical magnetic image to obtain the best fit of the images shown in (**a–d**). The Pearl lengths obtained are $\Lambda = 111$ **e**, 101 **f**, 30 **g**, and 12 **h** μm (**i–l**) Profile of the experimental data $B_z^{ac}(h, x)/x_{ac}$ (brown), and the calculated vortex (yellow). **i** Calculated vortex profile with $\Lambda = 94$ (blue), 128 (red) μm, which is a $\pm 15\%$ deviation from the best fit, all curves are multiplied by a factor 6 for clarity.

demonstrates that it is possible to distinguish variations in $\Lambda$ of the order of 15%, given our measurement sensitivity (Supplementary Note 3). This resolution is largely sufficient when compared to our claims. Another potential issue is the disorder and surface roughness, which might introduce possible limitation on the measured film thickness. To exclude this mechanism, cross-sectional STEM images were taken (Supplementary Fig. 1) to ensure the crystal quality and assess contamination for all thin films ($N \leq 14$). The flakes were found to be atomically flat without any trace of contamination over micron-size terraces where vortices were imaged.

## Discussion

The sharp jump of $\Lambda$ near $N = 6$ indicates a sharp reduction in the total superfluid density. NbSe$_2$ is known to be a two-band superconductor[26], with the larger energy band associated with the Nb orbitals, and the lower energy band with the Se orbitals. Hence a possible explanation for this reduction is the suppression of one of the bands with the thickness of the film[18,20,21,27,28]. Although that could explain the jump in $\Lambda$, it does not account for the saturation of $\Lambda$ in thinner films as the $1/d$ dependence should persist albeit with a different offset. Moreover, in NbSe$_2$, the spectral signature of the Se-derived band disappears

gradually at films in the $d \sim 10$ to 20 nm thickness range[21], well above the thickness of 3.5 nm ($N = 6$) where we observe the transition.

The distinctive behavior of $\Lambda$ presented in Fig. 4 suggests the presence of a phase transition wherein a partial disappearance of superconductivity occurs as the film thickness diminishes, thus leading to a notable jump in the Pearl length. At the same time, the saturation in the Pearl length implies the persistence of the superfluid density independent of the film thickness. This feature indicates that the remaining superconducting order parameter is confined in specific layers, and is therefore unaffected by the film thickness. Thus, the order parameter, which is suppressed for thinner films, prevails elsewhere for $N > 6$.

The observed jump in the Pearl length dependence as a function of the number of layers in the film, $\Lambda(N)$, can be naturally explained within the framework of the two-component Ginzburg-Landau description, in which $\varphi$ and $\psi$, represent superconducting order parameters belonging to different superconducting classes. In this description, the Pearl length is given by the sum of two contributions,

$$\frac{1}{\Lambda(N)} = N\frac{|\psi|^2}{\Lambda_b} + \frac{|\varphi|^2}{\Lambda_s} \tag{1}$$

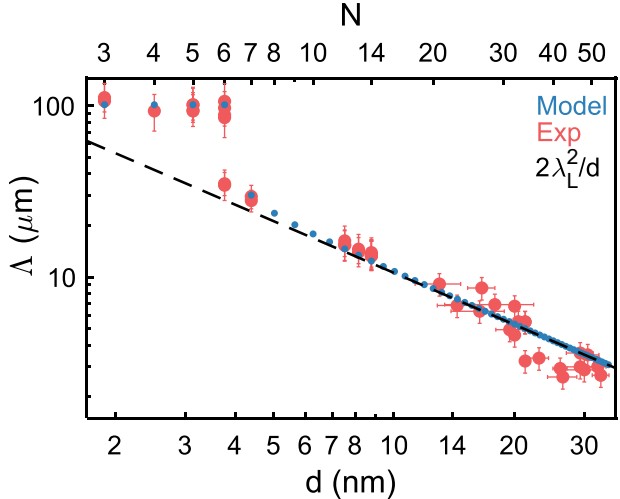

**Fig. 4 | Thickness dependence of the Pearl length.** Measured Pearl length Λ (red dots) obtained from SQUID-on-tip (SOT) images and the best fit to our phenomenological model (blue dots). The black dashed line represents Pearl length thickness dependence according to the Pearl model considering $\lambda_L = 230$ nm. The error bars represent the total uncertainty resulting from the systematical and statistical (one standard deviation) uncertainties as described in Supplementary Note 3 and Supplementary Figs. 4, 5. The sudden suppression of bulk superconductivity $\psi$ depicted in Fig. 1 is reflected by the sharp increase of Λ at N = 6 layers.

where $\Lambda_b$ is a length scale characterizing the bulk, while $\Lambda_s$ is associated with the surface. The order parameters $\varphi$ and $\psi$, are determined by minimization of the dimensionless energy density of an effectively two-dimensional system:

$$F_N = n^*|\varphi|^2(-2 + |\varphi|^2 + \mu|\psi|^2) + N|\psi|^2(-2 + |\psi|^2) \qquad (2)$$

where $n^*$ and $\mu$, are fitting parameters that control the competition between the two order parameters. The length scales, $\Lambda_b$ and $\Lambda_s$, are uniquely determined from the asymptotic limits of the Pearl length at large and small film thicknesses. They are found to be: $\Lambda_b = 2\lambda_L^2/a$ and $\Lambda_s = 0.6\Lambda_b$, where $\lambda_L = 230$ nm is the bulk London penetration depth, while $a = 0.63$ nm is the spacing between neighboring layers. The two other fitting parameters, $n^*$ and $\mu$, are chosen such that $\Lambda(N)$ follows the data points and the system undergoes a first-order phase transition at thickness of approximately $N \sim 6$ layers. This transition transforms the system from a phase where both $\psi$ and $\varphi$ are non-zero ($N \geq 7$) to a phase where $\psi$ vanishes while $\varphi$ remains non-zero ($N \leq 6$). The experimental data is perfectly described by $n^* = 6.5$ and $\mu = 1.9$, as shown in Fig. 4. In this model, the energy associated with the $\varphi$-component is independent of the number of layers, $N$, therefore it is associated with a contribution coming from the surface of the film (i.e., the first and last layers). Furthermore, the experimental data does not corroborate the presence of a term proportional to $\varphi^*\psi + \varphi\psi^*$ in the free energy. Hence, $\psi$ and $\varphi$ must belong to distinct irreducible representations of the system's superconducting symmetry group, linked to the point groups $D_{3h}$ and $D_{3d}$ for films with an odd or even number of layers, respectively. These irreducible representations are both one-dimensional; otherwise, the system should exhibit an additional type of symmetry breaking. The sensitivity of our measurements is insufficient for unveiling such a symmetry-breaking.

The phenomenological model proposed herein assumes the presence of a superconducting order parameter localized near the sample surface, distinct from that of the bulk. This order parameter may originate from various factors, such as a strong Rashba spin-orbit coupling at the surface or the presence of strain induced by lattice mismatch between hexagonal h-BN and NbSe$_2$. Both, Rashba-spin-orbit and strain gradient, induce an effective pseudomagnetic field

acting with opposite signs on electrons associated with distinct valleys[29]. The strain engendered by the lattice mismatch permeates into the bulk to a depth of approximately the effective lattice constant of the resulting moiré patterns. This depth is estimated to be on the order of 1 nm, significantly smaller than the sample thickness at which the transition manifests and thus consistent with our data. Additionally, it is noted that the coexistence of superconductivity with a different order parameter, such as charge density waves, does not alter the fundamental framework of our phenomenological model. However, it may influence its parameters and serve as a mediator for the interaction between bulk and surface superconducting orders.

The current study identifies a superfluid-density transition that has not been previously observed in transport or tunneling analyses[3,9,19–21]. Notably, previous tunneling investigations were conducted at temperatures of 1.2 K[3], 300 mK[20] and below 100 mK[9,21], whereas our experiment was conducted at $T = 4.2$ K. This suggests that the observed transition may be a distinct feature of intermediate temperature regime. While tunneling experiments have also been conducted at $T = 4.2$ K, these were limited to bulk crystals, where suppression of surface states is significant. It is also important to highlight that tunneling experiments using STM cannot fully encapsulate the sample, as the top surface must remain exposed. Our experimental study suggests that encapsulating the sample with hBN on both surfaces might be crucial to prevent degradation and enabling measurement of the intrinsic properties of the sample or to strain the NbSe$_2$ as discussed above. This limitation could be addressed through device-based tunneling experiments, where the top tunneling electrode is deposited on a thin hBN layer. Future SOT measurements at millikelvin temperatures will be necessary to directly compare these results with tunneling experiments.

One might expect that an abrupt change in the superfluid density as a function of film thickness would also manifest in the critical temperature. However, previous transport studies of the $T_c$ thickness dependence revealed only a small change between the bulk value and that observed in 5–6 layer devices[18]. Notably, the ultrathin NbSe$_2$ flakes in ref. 18 were exfoliated on SiO$_2$ and solely top-encapsulated, while the samples in our study are encapsulated with hBN on both sides. Moreover, assessing $T_c$ in two-dimensional superconductors through transport measurements faces several inherent limitations that are difficult to circumvent. Firstly, these measurements are highly sensitive to the nature of the vortex dynamics and pinning effects[30], which can potentially mask the observed transition, which is measured under equilibrium conditions. Secondly, transport measurements require metallic leads, which influence the superconducting state through the reverse proximity effect. Thirdly, $T_c$ measurements are sensitive to the order parameter only near the critical temperature, making it impossible to probe transitions occurring at intermediate temperatures as in our experiment.

Measurements of the in-plane critical magnetic field as a function of temperature could provide valuable evidence supporting our findings, provided orbital depairing effects play a significant role. This assumption is plausible, as the zero-temperature out-of-plane coherence length (approximately 2.3 nm[31]) is considerably larger than the interlayer spacing (0.63 nm). Under these conditions, two distinct transitions as a function of the in-plane magnetic field are anticipated: one corresponding to the bulk superconducting order parameter and another, at a higher field, associated with the surface superconducting order parameter.

Such experimental observations have indeed been reported in ref. 32, where a kink in the critical field versus temperature curve was identified. This kink is interpreted as a first-order transition into a Fulde-Ferrell-Larkin-Ovchinnikov (FFLO) superconducting state. Here, based on our data, we offer an alternative interpretation—namely that the observed kink is in fact the hallmark of the surface superconducting order parameter. Furthermore, consistent with our

theoretical explanation, the kink became progressively weaker as the film thickness increased. For films thinner than 5 nm−near the thickness at which we observed the transition−the critical field no longer exhibited the kink associated with the additional transition. These differing interpretations highlight the need for further investigations to distinguish between these scenarios or to reconcile them within a unified framework.

## Conclusions

In summary, the present study has unveiled a remarkable anomaly in the Pearl length dependence in the few-layer limit. This anomaly is interpreted as a phase transition in thin films of NbSe$_2$, wherein a reduction in film thickness triggers the suppression of bulk superconductivity, giving rise exclusively to surface superconductivity. Our approach provides invaluable insights into the underlying superconductive characteristics of the system. Specifically, irrespective of the theoretical framework, our experimental methodology serves as a powerful tool for detecting surface superconductivity and the concurrent suppression of bulk superconductivity.

The utility of Pearl-length measurements extends beyond this investigation, as they serve as a sensitive tool for probing the superconducting nature of thin films under various experimental conditions, such as elastic strain, electric field perturbations, or alterations in temperature. This versatility underscores the broader implications of our findings, advancing our understanding of superconductivity in diverse contexts and paving the way for further studies.

## Methods

### Sample fabrication

hBN was exfoliated onto 285 nm SiO$_2$/Si substrates, yielding flakes with thicknesses ranging from 5 to 15 nm for the upper layer and 10−20 nm for the lower hBN layer. To prepare NbSe$_2$ flakes, an initial exfoliation was carried out onto PDMS, followed by a transfer onto 285 nm SiO$_2$/Si substrates. This process resulted in the production of large, thin flakes displaying uniform steps. Flakes with consistent and sizable steps were identified utilizing optical microscopy, encompassing various thicknesses. Subsequently, a polycarbonate (PC) pickup technique was employed, as detailed in ref. 33. The pickup started with top hBN, followed by the NbSe$_2$ flake, and finally the lower hBN layer. The resultant structure was positioned onto a SiO$_2$ chip, near a predeposited gold heater used for SQUID-on-tip localization. Both the pickup procedure and the exfoliation of NbSe$_2$ were executed within an argon environment. This approach aimed to prevent degradation of the samples and maintain their integrity throughout the experimental process.

### SQUID-on-tip fabrication

The SOT was prepared using a self-aligned three-step process involving thermal deposition of Pb at cryogenic temperatures, as outlined in earlier works[16,17]. The SQUID loop employed in this work have a diameter of approximately 250 nm. This choice of a relatively larger diameter is strategic, as it provides the enhanced magnetic sensitivity required for measuring weak extended signals obtained by $\Lambda = 100\,\mu m$ vortices. Additionally, the larger diameter causes lower period of the quantum interference pattern of the SQUID critical current versus applied field. This enhances the zero-field magnetic sensitivity[16,17], which is essential for measurements at low vortex densities.

### SQUID-on-tip measurements

All measurements were carried out at a temperature of 4.2 K, which is below the critical temperature of 5.5 K[20] for NbSe$_2$ flakes with three layers or more. To initiate the experimental process, we applied small 0.3 mT magnetic fields to induce the creation of multiple vortices within the sample. Subsequently, we gradually reduced the magnetic field to a range of approximately −0.05 mT to 0.05 mT. Our primary objective during this phase was to identify a relatively isolated vortex positioned at a distance of 1 $\mu m$ or more from the edges of the uniform steps. Once such a vortex was located, we employed the Tuning fork (TF) to establish a distance akin to that of an AFM. Subsequently, images of these vortices were captured for further analysis.

### Sample characterization

The thicknesses of the NbSe$_2$ flakes were assessed utilizing Scanning Transmission Electron Microscopy (STEM). Specific areas of interest within the sample were carefully selected to encompass the approximate regions where the vortices were observed. These selected areas were subsequently fabricated into thin lamellas using a Focused Ion Beam (FIB) technique. These lamellas were then subjected to STEM imaging, with the outcomes presented in Supplementary Fig. 1. To ensure the integrity of the samples, thorough examinations for oxidation and degradation were conducted through Energy Dispersive X-ray Spectroscopy (EDS). The analysis confirmed an organized crystal structure, with no discernible signs of oxidation or degradation.

### Tip-to-sample distance measurement

The SOT is attached to a tuning fork excited electrically near its resonance frequency at $\sim$ 32 kHz. The quality factor of the tuning fork coupled with the SOT varies between 10000 and 30000 at 4 K with a 10 mbar helium gas pressure. The electrical signal is amplified at room temperature using a homemade operational amplifier. The amplified signal is fed to a nanonis lock-in amplifier in phase-locked loop mode. As the tip approaches within a few nm of the surface, a change in the phase is detected and the tip is retracted by a known safe distance. It was previously calibrated that once the phase variation is detected, the edge of the tip is within 1−2 nm from the sample surface. Thus, the sample-tip distance is set by the distance at which we retract the tip (typically around 300 nm).

## Data availability

The data that supports the findings of this study have been deposited in the GitHub database: https://github.com/QIL123/NbSe2_Thin_vortex, https://doi.org/10.5281/zenodo.14852386.

## Code availability

The MATLAB scripts that analyze the raw data and reproduce the figures appearing in this paper have been deposited in the GitHub database: https://github.com/QIL123/NbSe2_Thin_vortex, https://doi.org/10.5281/zenodo.14852386.

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

## Acknowledgements

We would like to thank Hermann Suderow, Avraham Klein, Isabel Guillamón, Maxim Khodas, Leonid Glazman, Boris Shapiro, Charis Quay Huei Li, Marco Aprili, Eli Zeldov, and Oded Millo for fruitful discussions. We thank Snir Gazit for the support in the data analysis. Y.A. acknowledges the support from the European Research Council (ERC) startup grant No. 802952 (STRONG) and consolidator grant No. 101124770 (MAJOR). H.S. acknowledges funding by Israel Science Foundation grant 164/23 and DFG Priority program grant 443404566. K.W. and T.T. acknowledge support from the JSPS KAKENHI (Grant Numbers 21H05233 and 23H02052) and World Premier International Research Center Initiative (WPI), MEXT, Japan.

## Author contributions

Y.A., H.S., O.A., T.D.F., and N.F. conceived the experiment. A.Z., N.F., T.R.D, E.G, and I.K fabricated the NbSe$_2$ devices. N.F. and T.D.F. conducted the scanning SOT measurements. O.A. and I.A. conceived the theoretical model. N.F., T.D.F, A.N, Y.Z, and A.G fabricated the SOT sensor. A.G. and T.D.F fabricated the Tuning forks. N.F., T.D.F., and Y.A. generated the numerical simulations. N.F., T.R.D., A.V., S.R., and T.D.F. characterized the NbSe$_2$ samples. N.F, T.D.F, and M.M. analyzed the data. Y.A. and A.N. constructed the scanning SOT microscope. M.E.H. Conceived the SOT readout electronics. K.W. and T.T. synthesized the hB.N. N.F., H.S., O.A., and Y.A. wrote the article with contributions from all authors.

## Competing interests

The authors declare no competing interests.
