## [Transparent Peer Review file · Nature Communications]

Anomalous Thickness Dependence of the Vortex Pearl Length in Few-Layer NbSe₂

Corresponding Author: Professor Yonathan Anahory

Version 0:

Reviewer comments:

Reviewer #1

(Remarks to the Author)

This work attempted to study the vortex in superconducting NbSe₂ using a scanning SQUID-on-tip (SOT) microscope. They found an unexpected trend in the Pearl length as a function of thickness and attributed this to order competition.

The technique of scanning SQUID-on-tip microscopy is undoubtedly still a relatively unique and powerful technique. The study of NbSe₂ materials is also relatively new and attractive.

However, I also find some major issues, and I list them below:

1. The most serious issue of the study is the lack of identifying the two orders. I think it is not an acceptable argument of order competition if the authors do not know what the two orders are. To be specific what are the two orders? The author could not provide any experimental evidence and the orders are largely speculated. If the authors meant the two orders are bulk superconductivity and surface superconductivity, what do they mean? Can the two types of superconductivity be called two types of orders? I cannot agree with this definition, and no property can be called an order.

Besides the unknown orders, are the authors absolutely sure that the abnormal trend is definitely only due to competition?

2. Is the Pearl length important? If important, why the authors did not even spend any sentence to explain what is Pearl length and why it is important?

3. The flow of the manuscript is very confusing at least a bit misleading. I do not think it is a good idea to list the conclusion of the Pearl length in Figure 1 and then explain how it is measured in Figures 2 and 3. In other words, if the technique of SOT coupled with a tuning fork is new to the audience, presenting the results without educating the audience on the basic principle will lead to total confusion to the audience when presenting Figure 1.

Hence, I am not able to support the study.

Reviewer #2

(Remarks to the Author)

The manuscript describes scanning SQUID experiments on few-layer superconducting NbSe₂. The authors systematically image vortices in their samples, determining their Pearl length as a function of samples thickness. The main result is that this length is found to have an unexpected dependence on sample thickness: below a thickness corresponding to 6 atomic layers of NbSe₂ the Pearl length sharply increases and stays fixed. The authors hypothesize that this is the result of surface superconductivity and the suppression of bulk superconductivity in NbSe₂ 6-layers-thick or less.

The work is original, interesting, and relevant to its field. The data analysis appears sound and the conclusions drawn from the data are well supported. The experimental methodology is also appropriate. In fact, this particular method using scanning SQUID-on-tip to measure Pearl-lengths may be useful in future investigations of superconducting thin films.

I therefore recommend the publication of this work after the authors address the following concerns:

1) On page 4, the authors write that they measure the tip-sample spacing h by "sensing the sample surface with the tuning fork as in AFM". Could the authors be more specific about how they determine h ? As they themselves write, their model requires this parameter and the Pearl length. If they do not get h right, this could have important consequences on the fit. What is the error on their determination of h ? How does this propagate into their determination of the Pearl length?

2) In the same paragraph the authors write that they take into account the finite spatial resolution of the SQUID by convolving the modelled image with a circle equal to the SQUID diameter of 250 nm. Do they also consider flux focusing effects? Could these be important? Do they consider the 100 nm shaking amplitude of the SQUID, which appears to be significant compared the SQUID diameter itself?

3) Minor point: Figures sa-d, and models e-h seem to this reader of limited value. The quality of the fits can be judged by the linecuts in i-l. Having seen the spatial derivative of a vortex in Fig 2e, it does not seem useful to replot a number of images which look exactly the same.

Version 1:

Reviewer comments:

Reviewer #1

(Remarks to the Author)

After reviewing the revision and response, I regret to inform you that I cannot support this publication, as the issues are too critical.

1.
In this study, the authors actually only observed an anomaly in the superfluid density, and then speculate that this might indicate a competition between two orders. While this reasoning is logical, I believe it is fundamentally flawed to overclaim that there are two orders are competing. If the paper's title is like "anomaly of superfluid density" that is fine. But, claiming "competition of two orders" as a conclusion is fundamentally wrong. Again, these authors only speculated there is a second order and failed to identify the second order. If they do not know what the second order is, how can they claim there is competition between two orders? For example, this is like a reviewer concluding a rejection of this paper by claiming there is a mistake, yet being unable to specify what that mistake is.

2.
I also still disagree with the authors' direct presentation of the Pearl length. The problem is not able the logical flow of writing. It is about fundamental scientific support of the figure 1's results. In this study, they used a method developed by themselves, which has never been quantitatively verified. Actually, the authors even admit errors in this technique in this study. Hence, presenting the Pearl length as a reliable conclusion in the beginning of the study seems like an attempt to manipulate the readers into believing the results are valid, when they have not been verified.

Reviewer #2

(Remarks to the Author)

The authors have satisfactorily responded to my concerns. I recommend publication.

Reviewer #3

(Remarks to the Author)

Fridman et al. measured the Pearl length of NbSe₂ flakes as a function of film thickness and discovered a sudden change of the Pearl length around 6 layers as well as an abnormal independence of the Pearl length on thickness N for $N < 6$. These findings were proposed to be indications of two competing superconducting orders: surface superconductivity and bulk superconductivity. The SQUID-on-tip technique is quite unique, and the data analysis is clear.

After cautiously going over the reports by the two reviewers and the responses by the authors, particularly the rebuttal to the first reviewer's criticism, I feel there's still room for further improvements that could help the authors convince more colleagues in the field to have faith in the two competing superconducting orders interpretation. The facts which disfavor the interpretation and were already mentioned by the authors, are that T_c determined by electric transport experiment manifests no such jump as the film gets thinner. Moreover, there is no other evidence, from transport or spectroscopy or other methods, supporting the existence of surface superconductivity. The jump of Pearl length is indirect evidence that could be open to many interpretations. Further investigations are needed to confirm this conjecture on which the importance of this work is established.

In fact, it is puzzling why the authors haven't shown the superfluid density measurements on their samples, which is what the scanning SQUID technique designed to do and famous for. In the absence of magnetic field, the superfluid density (or superfluid stiffness, which differs from the density by a factor of the effective mass), can be retrieved from the diamagnetic signals that is a direct characterization of superconductivity. If there exists only surface superconductivity for $N < 6$, then the superfluid density should be small and independent on N . And for $N > 6$, one would expect to observe two transitions in diamagnetic signal corresponding to surface and bulk superconductivity respectively. If the proposed conjecture is right, then there would be a sudden jump in superfluid density as N goes across 6 layers.

I strongly encourage the authors to consider the above experiment for the current evidence is too weak to support their conjecture. If the SQUID-on-tip instrument is lack of the precision or not suitable for whatever reason, it would be wise to use a normal scanning SQUID or collaborate with someone to provide this key information.

Version 2:

Reviewer comments:

Reviewer #1

(Remarks to the Author)

I feel that the authors have addressed my comments, and I am ok to support a publication.

Reviewer #3

(Remarks to the Author)

Please see the attached document.

Reviewer #1 (Remarks to the Author):

This work attempted to study the vortex in superconducting NbSe₂ using a scanning SQUID-on-tip (SOT) microscope. They found an unexpected trend in the Pearl length as a function of thickness and attributed this to order competition.

The technique of scanning SQUID-on-tip microscopy is undoubtedly still a relatively unique and powerful technique. The study of NbSe₂ materials is also relatively new and attractive. However, I also find some major issues, and I list them below:

1. The most serious issue of the study is the lack of identifying the two orders. I think it is not an acceptable argument of order competition if the authors do not know what the two orders are. To be specific what are the two orders? The author could not provide any experimental evidence and the orders are largely speculated. If the authors meant the two orders are bulk superconductivity and surface superconductivity, what do they mean? Can the two types of superconductivity be called two types of orders? I cannot agree with this definition, and no property can be called an order.

Besides the unknown orders, are the authors absolutely sure that the abnormal trend is definitely only due to competition?

We would like to address your concern regarding the identification of the two orders in our study, which you highlighted as a key issue. We recognize the importance of this question and provide the following clarification.

When we refer to "two orders" in our work, we are specifically discussing superconducting order parameters that correspond to different irreducible representations of the superconducting group. These representations differ depending on whether the system is in the D_{3h} symmetry (for odd-numbered layers) or D_{3d} symmetry (for even-numbered layers). One of these orders is reasonably understood: it is the bulk s-wave superconducting order, which gradually transforms into the Gorkov-Rashba superconducting order near the surface. This order mixes a singlet and triplet component due to the Rashba spin-orbit coupling that becomes significant at the surface. It belongs to the A₁ (or A_{1g}) irreducible representation, which is the most symmetric state.

As for the second superconducting order, we acknowledge that our experimental setup does not allow us to determine its precise nature. However, we have strong experimental evidence that points to the presence of a second competing order, reflected by a clear discontinuity in the superfluid density and the resulting jump in the Pearl length. Furthermore, the saturation of the superfluid density, as the number of layers falls below six, indicates that this second order parameter is confined primarily to the surface. While we can speculate on possible candidates for the second order, such as a different irreducible representation or an unconventional pairing symmetry, it would be premature to definitively state its nature without further experimental confirmation.

It is important to emphasize that the absence of a precise identification of the second order does not diminish the significance of our findings. The clear experimental evidence of competition between two distinct superconducting orders is, in itself, a critical result. Our intention in publishing this work is to bring attention to this phenomenon and to encourage further experimental investigations that could provide a more detailed characterization of the second order. As is often the case in scientific research, identifying such an intriguing effect is the first step, and subsequent work by us and others will be required to fully understand its nature.

We believe that this is a meaningful contribution to the field and that stimulating further research on this topic is a key part of the scientific process. While the nature of the second order remains an open question, we hope that our results will pave the way for future studies to resolve this issue.

2. Is the Pearl length important? If important, why the authors did not even spend any sentence to explain what is Pearl length and why it is important?

The Pearl length plays a critical role in thin film superconductors, replacing the London penetration depth that typically characterizes three-dimensional systems. It is a fundamental characterizer of superconductors, providing essential insights into the screening behavior and superfluid properties of the system. It serves as the analogous screening length in two-dimensional electrostatics. Unlike in three-dimensional samples, where charge is exponentially screened, thin films exhibit power-law screening, making the Pearl length a highly sensitive parameter for detecting variations in the two-dimensional superfluid density. This sensitivity is particularly important when studying phase transitions that involve changes in the superconducting order, as such transitions naturally alter the superfluid density.

One of the key innovations in our paper is the ability to measure the Pearl length with precision, despite its being much larger than the size of the superconducting flakes. By using a highly sensitive technique—measuring the derivative of the magnetic profile near a vortex with a SQUID-on-tip system coupled to a tuning fork—we were able to reliably extract the Pearl length. This technological advancement is essential to our results.

With this understanding in place, it becomes clear that the core of our work is effectively summarized in Fig. 1b, which presents the main findings regarding the Pearl length. Upon reflection, we recognize that our original explanation of the Pearl length's importance may not have been sufficiently clear, and we have revised the manuscript to provide a more detailed and accessible discussion of this crucial point. The third paragraph now reads as follows:

“Superconducting vortices are often exploited as local perturbations used to probe the properties of the order parameter.^{9–12} Vortices consist of a core where the superconductivity is locally suppressed on the scale of the coherence length ξ where the magnetic field can penetrate. This field is screened by the surrounding superconducting currents and results in magnetic flux quantization. In bulk superconductors, the magnetic field is screened exponentially with a characteristic scale of the London penetration depth λ_L . By contrast, for a film of thickness $d < \lambda_L$, screening is less effective and is governed by the Pearl length $\Lambda = 2\lambda_L^2/d$.^{13–15} In this limit, the magnetic field decays as $1/\Lambda r$ near the vortex core and Λ/r^3 for distances greater than Λ , where r is the distance from the vortex's center. Therefore, near the vortex core, there is no characteristic length scale for the magnetic field screening. In this work, we measure the Pearl length, which is a fundamental characterizer of superconductors that is highly sensitive to variation of the two-dimensional superfluid density. This sensitivity is particularly important when studying phase transitions that involve changes in the superconducting order, as such transitions naturally alter the superfluid density.”

3. The flow of the manuscript is very confusing at least a bit misleading. I do not think it is a good idea to list the conclusion of the Pearl length in Figure 1 and then explain how it is measured in Figures 2 and 3. In other words, if the technique of SOT coupled with a tuning fork is new to the audience, presenting the results without educating the audience on the basic principle will lead to total confusion to the audience when presenting Figure 1.

Hence, I am not able to support the study.

In structuring the manuscript, we adhere to two guiding principles: (1) Present the main result to the reader as early as possible, without disrupting the logical flow of the argument; and (2) Defer technical details to later sections. In many cases, background information must be established before presenting the main findings, often requiring multiple figures. However, in this case, the situation is different. Once the reader understands the key concept that the Pearl length is a direct measure of the superfluid density—and thus provides critical insight into the superconducting order parameter—the stage is set to present the main result without further delay.

Therefore, we believe it is appropriate to introduce the main result immediately in Fig. 1. This approach allows us to focus on the central finding early on and then proceed with a more detailed discussion of the technical aspects, such as the measurement technique, sample quality, and other relevant specifics.

As mentioned above, we acknowledge that our initial explanation of the Pearl length's importance may not have been sufficiently clear. We have made revisions to correct this issue and provide a stronger context for its significance. We remain confident that presenting the main result in Fig. 1 is the most effective way to communicate the findings to the reader.

Reviewer #2 (Remarks to the Author):

The manuscript describes scanning SQUID experiments on few-layer superconducting NbSe₂. The authors systematically image vortices in their samples, determining their Pearl length as a function of samples thickness. The main result is that this length is found to have an unexpected dependence on sample thickness: below a thickness corresponding to 6 atomic layers of NbSe₂ the Pearl length sharply increases and stays fixed. The authors hypothesize that this is the result of surface superconductivity and the suppression of bulk superconductivity in NbSe₂ 6-layers-thick or less.

The work is original, interesting, and relevant to its field. The data analysis appears sound and the conclusions draw from the data are well supported. The experimental methodology is also appropriate. In fact, this particular method using scanning SQUID-on-tip to measure Pearl-lengths may be useful in future investigations of superconducting thin films.

I therefore recommend the publication of this work after the authors address the following concerns:

We thank the reviewer for acknowledging the relevance of our work and for raising the important points regarding the data analysis and the measurements. We believe that our addressing these comments improved our manuscript.

1) On page 4, the authors write that they measure the tip-sample spacing h by "sensing the sample surface with he tuning fork as in AFM". Could the authors be more specific about how they determine h ? As they themselves write, their model requires this parameter and the Pearl length. If they do not get h right, this could have important consequences on the fit. What is the error on their determination of h ? How does this propagate into their determination of the Pearl length?

We thank the reviewer for highlighting this unclear point. We agree that the method of measuring the tip-sample distance was not adequately detailed. We added a section describing this technique in the methods (see below).

"The SOT is attached to a tuning fork excited electrically near its resonance frequency at ~ 32 kHz. The quality factor of the tuning fork coupled with the SOT varies between 10 000 and 30 000 at 4 K with a 10 mbar helium gas pressure. The electrical signal is amplified at room temperature using a homemade operational amplifier. The amplified signal is fed to a nanonis lock-in amplifier in phase-locked loop mode. As the tip approaches within a few nm of the surface, a change in the phase is detected and the tip is retracted by a known safe distance. It was previously calibrated that once the phase variation is detected, the edge of the tip is within 1-2 nm from the sample surface. Thus, the sample-tip distance is set by the distance at which we retract the tip (typically around 300 nm)."

The uncertainty on h was discussed in supplementary Note 3 but was not explicitly mentioned in the main text. Following the reviewer's comment, we explicitly mention the uncertainty on the measurement of h and how it propagates into the Pearl length uncertainty (see below). We emphasize that an uncontrolled uncertainty in the determination of h would translate into a systematic error in measuring the Pearl length. It cannot explain the essential features relevant to our conclusion, which are: (1) the atomically sharp jump in the Pearl length at $N = 6$ (2) the thickness-independent value of the Pearl length for $N \leq 6$. Moreover, we note that for thick samples, the measured Pearl length is consistent with $\lambda_L = 230$ nm, which is close to the bulk value taken from the literature $\lambda_L = 200$ nm. We are therefore confident that the height measurement is well calibrated within an uncertainty of 15 nm. Following the reviewer's comment, we added the following phrase to the main text.

"The uncertainty on the measurement of h is 15 nm, which propagates to an uncertainty in the Pearl length of 8%-10% (Supplementary Note 3)"

2) In the same paragraph the authors write that they take into account the finite spatial resolution of the SQUID by convolving the modelled image with a circle equal to the SQUID diameter of 250 nm. Do they also consider flux focusing effects? Could these be important? Do they consider the 100 nm shaking amplitude of the SQUID, which appears to be significant compared the SQUID diameter itself?

We thank the reviewer for raising these potential artifacts. The effect of flux focusing is expected to be small because all the superconducting layers are thinner than 25 nm, which is smaller than the London penetration depth $\lambda_L \approx 100$ nm in Pb. Moreover, the leads are nearly parallel to the out-of-plane direction (z-axis in Figure 1a), therefore, only a tiny part of the leads are near the loop. From previous experience, the diameter of the loop we measured with the SEM always closely matched the area obtained from the SQUID interference pattern (see Refs. 16,17 from the manuscript). It always gives a diameter smaller than the outer diameter but larger than the inner diameter, thus confirming the negligible influence of flux focusing.

To affect our measurement, the flux focusing effect should change the tip diameter by more than one pixel (60 nm), which was never observed. Finally, as stated in previous comment, flux focusing would generate a systematic error that cannot explain the critical aspect of our results. Moreover, comparing λ_L obtained for thick films suggest that this systematic error is smaller than 15%. In conclusion, we do not think flux focusing is a significant source of uncertainty. Following the reviewer's comment, the following statement was added to the Supplementary Material.

“3.1.3 Uncertainty on the tip diameter

The uncertainty of the tip diameter influences the measured Λ because the image resulting from the Pearl model is convoluted with the tip size to account for finite tip-size smoothing. We note that the convolution method is only sensitive to a change in diameter greater than the pixel size (60 nm). The tip diameter was determined by measuring the field period of the SQUID interference pattern ΔH_z . Knowing that one period corresponds to a change in flux corresponding to the flux quantum Φ_0 , we can state that $\pi r^2 \Delta H_z = \Phi_0$. The uncertainty of the field period is small (a few percent). In principle, other effects, such as flux focusing, could change the effective diameter. However, such an effect was never observed on a magnitude comparable with the pixel size (Ref. 16 and 17 of the main text). The negligible influence is most likely due to the thin film geometry found in the SOT. For those reasons, we consider the uncertainty on the tip diameter as negligible.”

Regarding the shaking amplitude of the SQUID, we note that 100 nm is smaller than the tip-sample distance ($h = 250$ to 400 nm), the SOT loop (250 nm), and the image size (≥ 2 μm). It averages over the two nearest pixels (60 nm). Our method relies on the linearity of the static magnetic field $B_z^{dc}(h, \mathbf{r})$ over the shaking amplitude length scale. The measured signal for vortices with a larger signal (small Pearl length) shows that this is a good approximation in our case. For example, see Figures 2c and e. Following the reviewer's comment, we added the following statement.

“Typically, we set $x_{ac} \lesssim 100$ nm, which yields a large signal while keeping the following approximation $B_z^{ac}(h, \mathbf{r}) \cong x_{ac} dB_z(h, \mathbf{r})/dx$ within our experimental uncertainty.”

3) Minor point: Figures sa-d, and models e-h seem to this reader of limited value. The quality of the fits can be judged by the linecuts in i-l. Having seen the spatial derivative of a vortex in Fig 2e, it does not seem useful to replot a number of images which look exactly the same.

We thank the reviewer for highlighting this redundancy in the figures. We believe showing the complete raw data is more convincing. For example, someone could have handpicked the most favorable line cut. Given that the fit considers the entire image, we felt it was important to show that data. Moreover, the small number of figures in this article, hence, we do not feel the paper is too crowded.

Reviewer #1 (Remarks to the Author):

After reviewing the revision and response, I regret to inform you that I cannot support this publication, as the issues are too critical.

1.

In this study, the authors actually only observed an anomaly in the superfluid density, and then speculate that this might indicate a competition between two orders. While this reasoning is logical, I believe it is fundamentally flawed to overclaim that there are two orders are competing. If the paper's title is like "anomaly of superfluid density" that is fine. But, claiming "competition of two orders" as a conclusion is fundamentally wrong. Again, these authors only speculated there is a second order and failed to identify the second order. If they do not know what the second order is, how can they claim there is competition between two orders? For example, this is like a reviewer concluding a rejection of this paper by claiming there is a mistake, yet being unable to specify what that mistake is.

We appreciate the referee's constructive criticism and, following his/her suggestion, have changed the article title to "Anomalous Thickness Dependence of the Vortex Pearl Length in Few-Layer NbSe₂". Additionally, we now present our theoretical explanation as a suggestive interpretation.

The last paragraph of the introduction now reads as follow:

"Our data presents the anticipated $1/d$ dependence for flakes of thicknesses $N \gtrsim 10$ layers. However, strikingly, Δ largely deviate from the expected $1/d$ dependence for thinner films. Such deviation has not been previously reported in NbSe₂ neither in transport¹⁸ nor in tunneling^{3,9,19-21} studies. We suggest that the sharp jump in Δ can be attributed to the competition between bulk and surface superconductivity."

2.

I also still disagree with the authors' direct presentation of the Pearl length. The problem is not able the logical flow of writing. It is about fundamental scientific support of the figure 1's results. In this study, they used a method developed by themselves, which has never been quantitatively verified. Actually, the authors even admit errors in this technique in this study. Hence, presenting the Pearl length as a reliable conclusion in the beginning of the study seems like an attempt to manipulate the readers into believing the results are valid, when they have not been verified.

We appreciate the referee's feedback and, in response, have moved the figure that displays the Pearl length versus thickness panel to the end of the article, where it is now designated as Figure 4. We refer to this figure only from the end of the results section.

Reviewer #2 (Remarks to the Author):

The authors have satisfactorily responded to my concerns. I recommend publication.

We are grateful to the reviewer for finding our manuscript suitable for publication.

Reviewer #3 (Remarks to the Author):

Fridman et al. measured the Pearl length of NbSe₂ flakes as a function of film thickness and discovered a sudden change of the Pearl length around 6 layers as well as an abnormal independence of the Pearl length on thickness N for $N < 6$. These findings were proposed to be indications of two competing superconducting orders: surface superconductivity and bulk superconductivity. The SQUID-on-tip technique is quite unique, and the data analysis is clear.

After cautiously going over the reports by the two reviewers and the responses by the authors, particularly the rebuttal to the first reviewer's criticism, I feel there's still room for further improvements that could help the authors convince more colleagues in the field to have faith in the two competing superconducting orders interpretation. The facts which disfavor the interpretation and were already mentioned by the authors, are that T_c determined by electric transport experiment manifests no such jump as the film gets thinner. Moreover, there is no other evidence, from transport or spectroscopy or other methods, supporting the existence of surface superconductivity. The jump of Pearl length is indirect evidence that could be open to many interpretations. Further investigations are needed to confirm this conjecture on which the importance of this work is established.

We thank the reviewer for thoroughly assessing our work. We understand that the main concern is our interpretation involving competing orders, and we acknowledge that such orders have not been identified in previous studies of this material. In the paragraph before the conclusion, we discuss how other measurements might have missed the order competition. However, in response to this criticism, we have refocused the article by changing the title and presenting the theoretical explanation as a suggestive interpretation (please see our response to Reviewer 1).

In fact, it is puzzling why the authors haven't shown the superfluid density measurements on their samples, which is what the scanning SQUID technique designed to do and famous for. In the absence of magnetic field, the superfluid density (or superfluid stiffness, which differs from the density by a factor of the effective mass), can be retrieved from the diamagnetic signals that is a direct characterization of superconductivity. If there exists only surface superconductivity for $N < 6$, then the superfluid density should be small and independent on N . And for $N > 6$, one would expect to observe two transitions in diamagnetic signal corresponding to surface and bulk superconductivity respectively. If the proposed conjecture is right, then there would be a sudden jump in superfluid density as N goes across 6 layers.

I strongly encourage the authors to consider the above experiment for the current evidence is too weak to support their conjecture. If the SQUID-on-tip instrument is lack of the precision or not suitable for whatever reason, it would be wise to use a normal scanning SQUID or collaborate with someone to provide this key information.

We appreciate the reviewer's suggestion to conduct an alternative measurement of the superfluid density using a scanning SQUID. While this approach is indeed a logical consideration, we believe it would not provide the desired confirmation for the following reasons:

1. Our method and the "diamagnetic signals" methods are equivalent. Fundamentally, both methods measures the Pearl length. Therefore, it would not introduce new information beyond what is already available through our current methodology.
2. It is practically impossible to measure the Pearl length with sufficient accuracy using the diamagnetic signal. Given the maximum lateral dimension of NbSe₂ flakes with one thickness ($< 10 \mu\text{m}$) and the size of the SQUID sensor—combined with the fact that the Pearl length is significantly larger than both.

We elaborate on these points in detail below.

Redundancy of measurement

It is well established in the scanning SQUID microscopy community that there are two primary methods for measuring the superfluid density with this technique: one involves susceptibility measurements of the diamagnetic response, and the other entails measuring the spatial decay of the magnetic field around a vortex.

Notably, calculating the superfluid density from susceptibility data requires the use of a model, such as the one described by Kogan [Phys. Rev. B **68**, 104511 (2003)], to extract the Pearl length. This method is summarized in the sixth paragraph of the introduction of Bert *et al.* [Phys. Rev. B **86**, 060503(R) (2012)].

Given that both methods ultimately rely on determining the Pearl length and hence fundamentally equivalent, performing a susceptibility measurement would not provide additional insights beyond what we have already obtained. Therefore, we believe that our current approach is appropriate and sufficient for the purposes of our study.

Practical limitations

The susceptibility method has indeed been used in the past to measure significantly larger Pearl lengths, such as $\Lambda \approx 1\text{mm}$, which might seem more challenging than our measurement of $\Lambda \approx 100\ \mu\text{m}$. However, this was achieved using a SQUID loop of $5.2\ \mu\text{m}$ on a sample that was several millimeters in size, as demonstrated by Kirtley *et al.* [Phys. Rev. B **85**, 224518 (2012)]. In our case, the much smaller dimensions of both the SQUID sensor and the sample make the susceptibility method impractical as we explain below.

Our measurement technique is unique in that it allows us to accurately extract the Pearl length with low error bars (within 20 %), even when the Pearl length is an order of magnitude larger than the flake size. This is achieved by coupling the SQUID sensor to a tuning fork, which enables us to measure the spatial gradients of the magnetic field in the immediate vicinity of a vortex. As detailed in the Supplementary Note 1 (see Eq. (11,a-c)), despite the presence of strong screening currents due to image vortices—because the Pearl length is much larger than the sample size—the error in our measurement remains of the order of $(h/L)^2$, where h is the distance between the tip and the sample, and L is the distance to the near sample edge.

In essence, while the absolute value of the magnetic field can vary significantly due to edge currents (comparable to the value in an infinite system), the gradient of the magnetic field we measure still reflects the true behavior of an effectively infinite sample. This allows us to obtain reliable data even in small samples.

On the other hand, measuring the magnetic susceptibility with a larger SQUID sensor on a sample whose size is much smaller than the Pearl length is expected to result in a relative error of approximately 100%. Such measurements are only reliable if the sample size is much larger than the Pearl length (i.e. samples of size of order of a millimeter and coated with hBN to mitigate surface effects), or if the boundary effects can be precisely calculated—a formidable task. Therefore, this alternative measurement presents significant technical challenges in both the experimental setup and sample fabrication.

To provide a concrete example, we refer to the recent work by Jarjour *et al.* [Nat. Commun. **14**, 2055 (2023)], where they measured the superfluid density using the diamagnetic signal on thin flakes. In that study, the samples were considerably larger (15 to 20 micrometers in diameter) because MoS₂ is easier to exfoliate into large, atomically flat terraces. Notably, the systematic uncertainty in the superfluid density was a factor of 2 to 4. Thus, while this technique is suitable for detecting changes in the superfluid density as a function of a tunable parameter—such as gate voltage—when the sample geometry is fixed, it cannot be effectively used to compare different samples measured in separate cooldowns, as in our study. The substantial uncertainty factor of 2 to 4 is too large to reach reliable conclusions in such comparisons.

Moreover, the etching technique employed in that study does not allow for coating with hexagonal boron nitride (hBN), which we find absolutely necessary for thin flakes of NbSe₂. The hBN coating is critical in our experiments to prevent sample degradation caused by oxidation.

Given these considerations, we believe that our current methodology is the most appropriate for this study, providing accurate and reliable results under the given experimental constraints.

Bibliography

- Kogan, V. G. "Meissner response of anisotropic superconductors." *Physical Review B* **68** (2003) 104511.
- Bert, Julie A., et al. "Gate-tuned superfluid density at the superconducting LaAlO₃/SrTiO₃ interface." *Physical Review B—Condensed Matter and Materials Physics* **86** (2012) 060503
- Jarjour, A., Ferguson, G.M., Schaefer, B.T. *et al.* "Superfluid response of an atomically thin gate-tuned van der Waals superconductor." *Nature Communications* **14** (2023) 2055
- Kirtley, J. R., et al. "Scanning SQUID susceptometry of a paramagnetic superconductor." *Physical Review B* **85** (2012) 224518

The authors gave an explanation why they thought an alternative measurement on the diamagnetic signal is unnecessary and the technical challenges in applying the diamagnetic measurements on the flake sample. However, as pointed out in my last report, the main concern of the work is whether the claimed sudden jump of the Pearl length is solid, and the interpretation based on two competing orders is sound.

We thank the reviewer for their feedback and for emphasizing the importance of supporting our interpretation. We acknowledge the concern regarding the robustness of the observed sudden jump in the Pearl length and the interpretation involving two competing orders. In response, we have further moderated the strength of our claim in this revision, expanding on the adjustments made in the previous round. Additionally, following Referee 3's suggestion, we now highlight our experimental findings from a different perspective (see Point 2 below), providing a more nuanced and comprehensive discussion of the results.

Regarding the jump in the Pearl length, we carefully revisited the extensive literature on NbSe₂ in response to the reviewer's comment. Through this deeper investigation, we identify independent evidence that we believe supports our observation. We are confident that this additional support effectively resolves the concern (see Point 2).

1. As referee 1 has pointed out, the effectiveness of the method employed by the authors to determine the Pearl length needs verification. Thus, an independent alternative diamagnetic measurement is not redundant but a strong support for confirmation of the claimed effect.

First, we would like to emphasize that following the revisions made in the previous round, Referee 1 and 2 have accepted the manuscript. Regarding the current criticism, it is important to recognize that many significant experimental findings in the literature have been validated using a single experimental technique. Referee 3 has not provided a clear scientific rationale to question the validity of our experimental approach. We find it unjustified to dismiss our results without providing substantial arguments specifically challenging the experimental method we employed. While proposing the development of a new experimental setup may seem straightforward in theory, its practical realization would be a considerable undertaking, potentially requiring several years (see further details below).

2. A transition from surface superconductivity to bulk superconductivity as the thickness increases should have observable consequences other than the Pearl length. The superconducting temperature T_c with no sign of discontinuity in fact is against the proposed interpretation. It is worth checking whether other quantities, e.g. critical magnetic field, critical current, or superconducting gap, show any discontinuity as a function of thickness. Without further evidence, it is hard to decide if this work can survive the scrutiny once more transport or spectroscopic evidence emerge.

First, we would like to reiterate that the revised manuscript now focuses on the distinct behavior of the Pearl length as a function of film thickness rather than its interpretation. However, regarding our interpretation, there are several reasons why the discontinuity in the Pearl length may not appear in T_c and superconducting gap measurements. Following the referee's comment, we have expanded the final paragraph of the discussion to address these points in detail.

Specifically, for T_c transport measurements, several factors could obscure the observed phenomenon: Firstly, transport-based T_c measurements are highly sensitive to vortex motion and pinning effects, which can mask subtle transitions in the superconducting state. Secondly, metallic contacts introduced during device fabrication can perturb the superconducting state through the reverse proximity effect. Thirdly, constructing a device with only a few layers while encapsulating it with hBN on both sides is technically demanding. Fourthly, T_c reflects only the behavior at the transition temperature, whereas the superconducting order parameter continues to evolve well below T_c . Therefore, transport measurements are inherently invasive and do not fully capture changes in the superfluid density below the transition.

Regarding tunneling spectroscopy, achieving higher energy resolution requires measurements at temperatures lower than 4.2 K. However, our study was limited to 4.2 K due to constraints of the SQUID-on-tip system. Lowering the measurement temperature for a direct comparison between Pearl length data and tunneling spectroscopy would be a substantial effort, likely requiring several years. Additionally, tunneling spectroscopy primarily probes surface layers, making it unsuitable for examining deeper regions. In the case of STM measurements, encapsulating the NbSe₂ sample from both sides is impossible since the STM tip must directly access the top layer.

Nevertheless, upon further consideration of the referee's feedback, we identified in-plane critical magnetic field measurements as a possible method to support our findings. Assuming that orbital depairing effects are significant [a reasonable assumption given that the zero-temperature out-of-plane coherence length (2.3 nm) is much larger than the interlayer spacing (0.63 nm)], two distinct transitions in the in-plane magnetic field are expected: one corresponding to the bulk superconducting order parameter and another, at a higher field, related to the surface superconducting order parameter. This behavior has been experimentally observed in Ref. 32 (now cited in the manuscript), where a kink in the critical field versus temperature curve was reported (see below, panels h & i from that reference). We interpret this kink as a signature of the surface superconducting order parameter. Notably, the kink diminishes as the film thickness increases, aligning with our theoretical framework. For films thinner than 5 nm—near the thickness where we observed the transition—the critical field no longer displayed the kink. While we observed the transition at 4 nm rather than 5 nm, the discrepancy could stem from different thickness measurement techniques (AFM in Ref. 32 versus cross-sectional TEM in our study).

The authors of Ref. 32 attribute the kink to a first-order phase transition into the FFLO superconducting state. However, these two interpretations are not necessarily mutually exclusive, and future research may develop a theoretical framework that unifies both explanations.

In response to the referee's comment, we have revised the discussion section to explain in greater detail why our findings are not evident in T_c and STM measurements. Additionally, we now discuss the results of Ref. 32 and present their model as an alternative perspective. We believe this additional analysis and supporting evidence substantially strengthen and corroborate our findings.

Revised discussion

The current study identifies a superfluid-density transition that has not been previously observed in transport or tunneling analyses.^{3,9,19–21} Notably, previous tunneling investigations were conducted at temperatures of 1.2 K,³ 300 mK²⁰ and below 100 mK,^{9,21} whereas our experiment was conducted at $T = 4.2$ K. This suggests that the observed transition may be a distinct feature of intermediate temperature regime. While tunneling experiments have also been conducted at $T = 4.2$ K, these were limited to bulk crystals, where suppression of surface states is significant. It is also important to highlight that tunneling experiments using STM cannot fully encapsulate the sample, as the top surface must remain exposed. Our experimental study suggests that encapsulating the sample with hBN on both surfaces might be crucial to prevent degradation and enabling measurement of the intrinsic properties of the sample or to strain the NbSe₂ as discussed above. This limitation could be addressed through device-based tunneling experiments, where the top tunneling electrode is deposited on a thin hBN layer. Future SOT measurements at millikelvin temperatures will be necessary to directly compare these results with tunneling experiments.

One might expect that an abrupt change in the superfluid density as a function of film thickness would also manifest in the critical temperature. However, previous transport studies of the T_c thickness dependence revealed only a small change between the bulk value and that observed in 5–6 layer devices.¹⁸ Notably, the ultrathin NbSe₂ flakes in Ref. 18 were exfoliated on SiO₂ and solely top-encapsulated, while the samples in our study are encapsulated with hBN on both sides.

Moreover, assessing T_c in two-dimensional superconductors through transport measurements faces several inherent limitations that are difficult to circumvent. Firstly, these measurements are highly sensitive to the nature of the vortex dynamics and pinning effects³⁰, which can potentially mask the observed transition, which is measured under equilibrium conditions. Secondly, transport measurements require metallic leads, which influence the superconducting state through the reverse proximity effect. Thirdly, T_c measurements are sensitive to the order parameter only near the critical temperature, making it impossible to probe transitions occurring at intermediate temperatures as in our experiment.

Measurements of the in-plane critical magnetic field as a function of temperature could provide valuable evidence supporting our findings, provided orbital depairing effects play a significant role. This assumption is plausible, as the zero-temperature out-of-plane coherence length (approximately 2.3 nm ³¹) is considerably larger than the interlayer spacing (0.63 nm). Under these conditions, two distinct transitions as a function of the in-plane magnetic field are anticipated: one corresponding to the bulk superconducting order parameter and another, at a higher field, associated with the surface superconducting order parameter.

Such experimental observations have indeed been reported in Ref. 32, where a kink in the critical field versus temperature curve was identified. This kink is interpreted as a first-order transition into a Fulde-Ferrell-Larkin-Ovchinnikov (FFLO) superconducting state. Here, based on our data, we offer an alternative interpretation – namely that the observed kink is in fact the hallmark of the surface superconducting order parameter. Furthermore, consistent with our theoretical explanation, the kink became progressively weaker as the film thickness increased. For films thinner than 5 nm —near the thickness at which we observed the transition—the critical field no longer exhibited the kink associated with the additional transition. These differing interpretations highlight the need for further investigations to distinguish between these scenarios or to reconcile them within a unified framework.

The authors gave an explanation why they thought an alternative measurement on the diamagnetic signal is unnecessary and the technical challenges in applying the diamagnetic measurements on the flake sample. However, as pointed out in my last report, the main concern of the work is whether the claimed sudden jump of the Pearl length is solid, and the interpretation based on two competing orders is sound.

1. As referee 1 has pointed out, the effectiveness of the method employed by the authors to determine the Pearl length needs verification. Thus, an independent alternative diamagnetic measurement is not redundant but a strong support for confirmation of the claimed effect.
2. A transition from surface superconductivity to bulk superconductivity as the thickness increases should have observable consequences other than the Pearl length. The superconducting temperature T_c with no sign of discontinuity in fact is against the proposed interpretation. It is worth checking whether other quantities, e.g. critical magnetic field, critical current, or superconducting gap, show any discontinuity as a function of thickness. Without further evidence, it is hard to decide if this work can survive the scrutiny once more transport or spectroscopic evidence emerge.

Based on these reasons, I do not believe that the work has met the high standards set by Nature Communications.